# What Drives the Morphological Traits of Stress-Tolerant Plant *Cynodon dactylon* in a Riparian Zone of the Three Gorges Reservoir, China

Xiaolong Li [1,2], Shanze Li [3], Yawei Xie [4], Zehui Wei [1] and Zilong Li [1,*]

1    Ocean College, Zhejiang University, Zhoushan 316021, China; lixiaolong@mwr.gov.cn (X.L.); 22134120@zju.edu.cn (Z.W.)
2    Ministry of Water Resources of the People's Republic of China, Beijing 100054, China
3    State Key Laboratory of Simulation and Regulation of Water Cycle in River Basin, China Institute of Water Resources and Hydropower Research, Beijing 100038, China; lishanze@126.com
4    Nanjiang Hydrogeology Brigade of Chongqing Geological Survey Bureau, Chongqing 401121, China; xieyawei-2@163.com
*    Correspondence: zilongli@zju.edu.cn; Tel.: +86-135-8873-6851

**Abstract:** The cyclical process of water storage and recession in the regular operation of the Three Gorges Reservoir creates a unique habitat stress that alters the structural and functional attributes of vegetation ecology within the riparian zone. The stress-tolerant plant *Cynodon dactylon* (L.) Pers is the dominant plant species in the riparian zone of the Three Gorges Reservoir. In this study, the riparian zone of the Daning River, a tributary located in the center of the Three Gorges Reservoir, was selected as our study area. To identify the drivers of the morphological traits of *C. dactylon* in the riparian zone of Daning River, we examined plant biomass and plant characteristics across different elevation gradients, with reference to abiotic factors to determine the distribution patterns of plant morphological traits. Results indicated that in the two main soil types of the riparian zone, plant biomass showed a consistent trend along the elevation gradient, with a "middle-height expansion" pattern; biomass increased and then decreased with rising water levels. Plant biomass positively correlated with soil total nitrogen and negatively correlated with soil pH, electrical conductivity, and total phosphorus. *C. dactylon* adapted to prolonged flooding in the riparian zone by having a significant negative correlation between plant height and erect stem length with soil moisture content to facilitate root respiration.

**Keywords:** Three Gorges Reservoir; riparian zone; soil; plant morphological traits; *Cynodon dactylon*





## 1. Introduction

The riparian zone of a reservoir, also known as the drawdown zone or disturbance zone, is a transitional ecosystem that alternately experiences submergence and exposure due to periodic fluctuations in water level [1]. These fluctuations can result from natural hydrological changes, such as seasonal water level fluctuations or anthropogenic manipulations, primarily due to cyclical water storage and discharge operations. Additionally, specific climatic events such as droughts can cause drawdown in reservoirs.

The Three Gorges Project is key for national infrastructure and a critical node in the water network of China. The Three Gorges Reservoir serves as an important strategic freshwater resource reserve for the nation and is a crucial ecological barrier in the upper reaches of the Yangtze River [2]. In 2010, the project achieved its experimental water storage target of 175 m for the first time. Since then, to maximize flood control benefits, the Three Gorges Reservoir has adopted a "winter storage and summer discharge" regulatory approach. The reservoir operates at a lower water level from April to October and at a higher water level from October to the following April. This unique operational scheduling

forms a unique fluctuating environment in the riparian zone with a drop range of 30 m, between 145 m and 175 m [1].

The cyclic water impoundment and release during normal operation of the Three Gorges Reservoir creates a unique form of habitat stress, altering the structure and function of the ecological pattern of vegetation in the riparian zone. In the cross-section of the riparian zone at different elevations, the ecological pattern of the vegetation exhibits notable spatial heterogeneity [3]. The biodiversity in the riparian zone is significantly affected by hydrological disturbances. Large-scale, unnatural water level fluctuations severely disrupt the original distribution of plant communities, resulting in a reverse succession in the riparian zone [4–6]. After the water impoundment operation of the Three Gorges Reservoir in 2003, the riparian zone gradually formed a vegetation succession pattern dominated by a few species including the plant *Cynodon dactylon* (L.) Pers, accompanied by *Atractylodes macrocephala* Koidz. and *Homalomena aromatica* Gagnep [1,7]. Since *C. dactylon* is the most dominant plant species, covering almost all the riparian zones of the Three Gorges Reservoir, we selected it as our study object. The plant *C. dactylon* shows significant resistance to flooding and nutrient-poor conditions [8,9]. This might have significant effects on the material conversion and ecological effects of element cycling in the riparian zone [10]. During the flooding period, the submerged plants will release several nutrients into the water, resulting in water eutrophication [11]. During the water-level drawdown period, plants are in their growing season.

The growth behavior and life history strategies of plants at each stage of their growth and development are closely related to the morphological traits of plants and can determine the distribution pattern and population behavior of plant species in the habitat to a certain extent. The adaptability of species is developed in the process of species evolution, which is manifested in morphology. Different morphological growth strategies are adopted by plants to maximize the fitness of species at certain stages. Plant morphological traits play an important role in studying biogeography in many ecosystems such as forestry [12] and agricultural [13] ecosystems. Under drought stress, plants can obtain the water content required for normal growth and development through the plasticity of morphological and structural characteristics, and adapt to the stressed environment by changing their morphogenesis. However, the morphological traits of plants in the riparian zone of reservoirs have received little attention. Differences in abiotic factors caused by climate change and geographical location differences will lead to different morphological characteristics of vegetation [14,15].

To maintain the *C. dactylon* community, it is important to address the morphological traits of this plant across a gradient of elevations, as well as its relationships with regional abiotic factors in the riparian zones. Hence, in this study, we tested the hypotheses that (1) *C. dactylon* morphological traits varied across elevations between two types of soil (yellow loam and purple soil) and (2) *C. dactylon* morphological traits varied with abiotic drivers.

## 2. Methods

### 2.1. Study Area

Our study area was located in a riparian zone (31.2813° N; 109.8179° E) of the Daning River (Figure 1). The Daning River is a tributary on the left bank of the Three Gorges Reservoir area. The region has a subtropical monsoon climate. The average annual temperature is 19.8 °C, with an average annual precipitation over 1000 mm. The Three Gorges Reservoir has a water level drop of 30 m. The types of riparian zones mainly include rocky bank slopes and gentle soil slopes. Our study was focused on the soil bank. Purple soil and yellow loam were the main soil types in the riparian zone of the Three Gorges Reservoir [16].

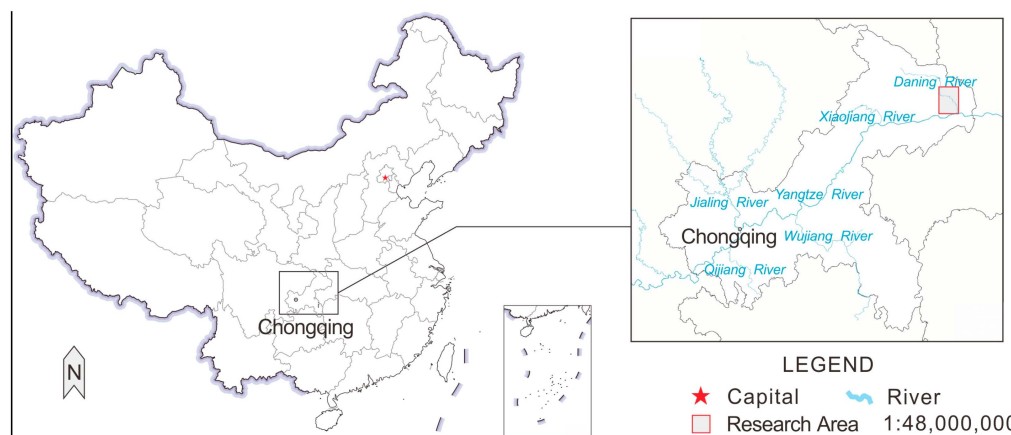

**Figure 1.** Study area.

### 2.2. Sample Collection and Testing

To give a comprehensive comparison of plant morphological traits between two types of soil, yellow loam and purple soil, across different elevations in a riparian zone of the Three Gorges Reservoir, China, we located three study sites in the riparian zones that are constituted of yellow loam and three study sites in the riparian zones that are constituted of purple soil. Thus, six riparian zones were selected. In each of the riparian zones, seven transects were selected from 145 m to 175 m (Figure 2). In each transect, we randomly set up three sampling quadrats as three replications of the plant community. At each sampling quadrat (1 m × 1 m), we recorded plant species and collected all the plant biomass. Plant biomass was dried in an oven (60 °C, 72 h), and weighed. We randomly selected six individuals of *C. dactylon* in each quadrat and recorded their erect stem length, number of ramets, average erect stem internode length, shoot height, first creeping stem, and root length.

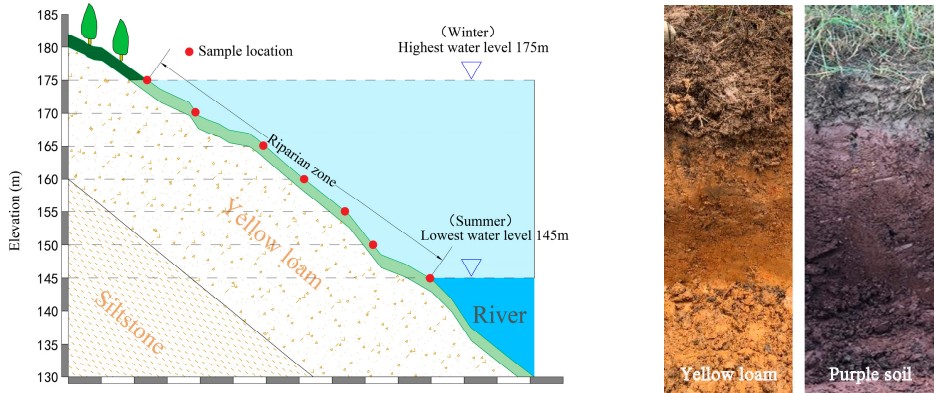

**Figure 2.** Schematic diagram of sampling locations in the Three Gorges Reservoir area (**Left**); photos of yellow loam and purple soil profiles (**Right**).

We examined environmental factors that might affect *C. dactylon* morphology, including soil types (yellow loam and purple soil), pH, electrical conductivity, total phosphorus, total nitrogen, moisture content, and temperature, and the duration of flooding. Three topsoil cores with a depth of 5 cm and a diameter of 5.05 cm were collected at each sampling point to obtain soil abiotic factors. Soil moisture content was determined by weighing wet soil cores and re-weighing them after drying for 48 h at 60 °C. Soil pH was analyzed by measuring the resulting supernatant of dry soil with deionized water at 1:5 *w/v*. Soil total nitrogen and total phosphorus were measured with a continuous flow analysis instrument [17]. The water level data were obtained from the Yangtze River Hydrology Bureau.

### 2.3. Data Analysis and Processing

We correlated yellow loam and purple soil with plant *C. dactylon* morphological traits (erect stem length, number of ramets, average erect stem internode length, shoot height, first creeping stem, and root length) using non-linear regression. The comparison between plant biomass in two soil types of riparian zone was carried out by using analysis of variance (ANOVA). We used Spearman's rank correlation analysis to examine potential relationships between plant morphology and soil abiotic factors (pH, electrical conductivity, total phosphorus, total nitrogen, moisture content, temperature, and duration of flooding). All statistical analyses were performed using SPSS 22 software (SPSS Inc., Chicago, IL, USA), where $p < 0.05$ was considered statistically significant [18].

## 3. Results

### 3.1. Dynamic Changes in Water Level in the Three Gorges Reservoir

The Three Gorges Reservoir has adopted a "winter storage and summer discharge" regulation mode. The water level slowly rises from 145 m to 175 m beginning at the end of September each year. The rising process generally takes about 60 days, during which time the riparian zone and its vegetation are gradually submerged. The high water level of 170~175 m then persists for approximately 60 days. Water discharge begins in April of the following year, and the water level gradually drops from 175 m to 145 m, a process that typically takes 200 days. During this time, the riparian zone and its vegetation gradually emerge from the water. A low water level of 145~150 m persists for approximately 90 days. The variations in water level in the Three Gorges Reservoir riparian zone over the past three years are shown in Figure 3.

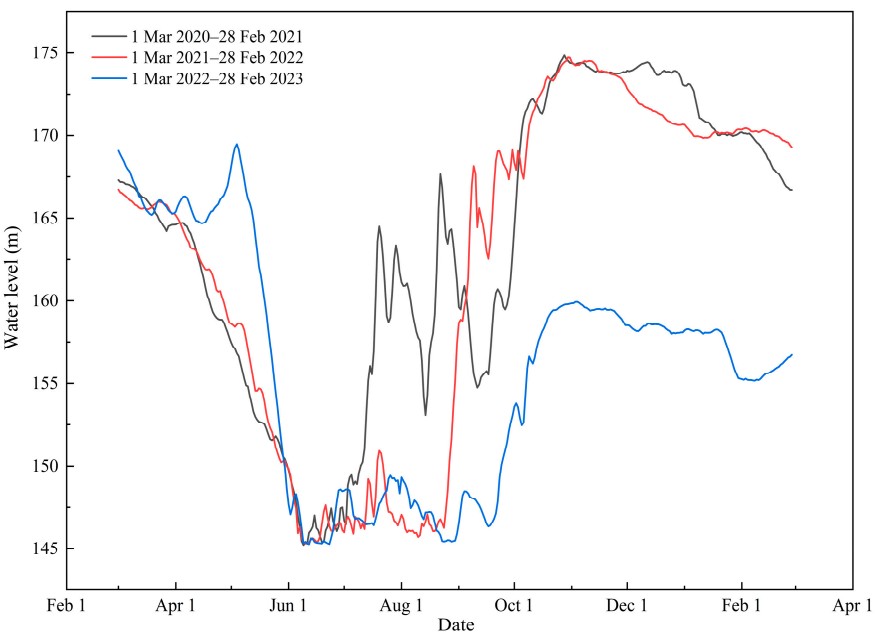

**Figure 3.** Variations of water level in the Three Gorges Reservoir.

### 3.2. Characteristics of Soil Type and Plant Biomass Distribution in the Riparian Zone

Through our survey, we found that the purple soil and yellow loam were the main soil types in the Daning River riparian zones. The proportions of clay and silt in yellow loam and purple soil reached as high as 66.44% and 56.4%, respectively, showing fine texture. As the drying time increases, the total nitrogen and phosphorus content in the soil gradually decreases.

The plant biomass at different elevations in the riparian zone under the two soil types is shown in Figure 4. This indicated significant differences in the plant biomass at different elevations in the yellow loam and purple soil riparian zones (F = 6.159, $p < 0.001$), but the

trend of plant biomass changes with water level elevation is consistent. That is, from an elevation of 145 m to 160 m, plant biomass increased with elevation. The plant biomass in yellow loam peaked at an elevation of 160 m to approximately 1395.40 g/cm$^2$, whereas the biomass in purple soil peaked at an elevation of 155 m to approximately 1291.17 g/cm$^2$. From an elevation of 165 m to 180 m, plant biomass showed a trend of decreasing.

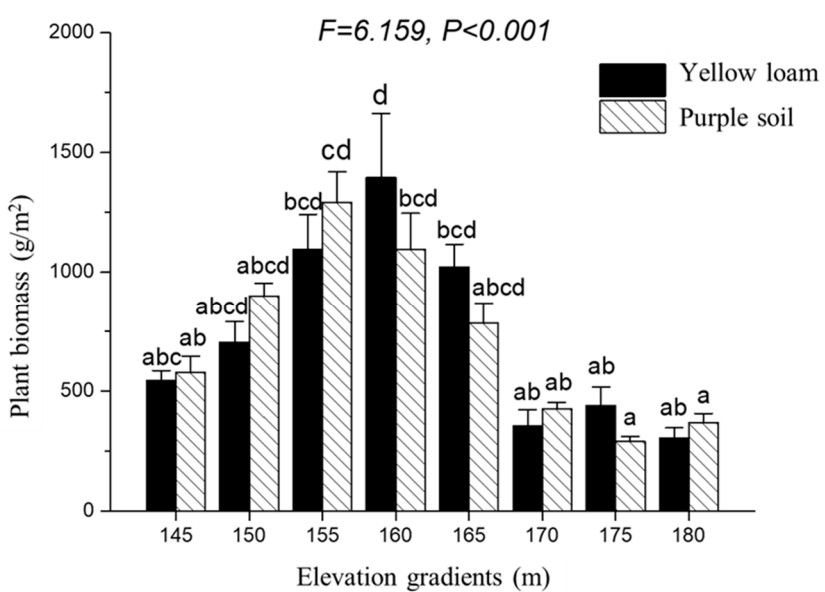

**Figure 4.** Distribution of plant biomass along different elevation gradients in the riparian zone of two soil types. Data are shown as means ± SE (*N* = 54). All ANOVA tests were significant (*p* < 0.05 in each case). The letter above each bar represents the results of post hoc Tukey's HSD test; bars sharing the same letter are not significantly different from one another.

*3.3. Plant Community and Morphological Traits of C. dactylon under Different Soil Types and Elevations*

The investigation results revealed that the dominant plant species within the 145 m to 165 m elevation range in the riparian zone of the Daning River were primarily *C. dactylon*, occasionally accompanied by *A. macrocephala* and *H. aromatica*. At elevations of 170 m and above, higher plant diversity was observed, including species such as *Daucus carota* L., *Ambrosia artemisiifolia* L., *Conyza canadensis* (L.) Cronq., *Digitaria sanguinalis* (L.) Scop., *Bidens frondosa* L., *Vitex negundo* L., *Melilotus officinalis* (L.) Desr., *Beckmannia syzigachne* (Steud.) Fern., and *Setaria viridis* (L.) Beauv. These species are mainly annual and perennial herbaceous plants. At higher elevations, shrubs, trees, and farmland were also present.

Within the elevation range of 145 m to 165 m in the riparian zone, the predominance of *C. dactylon* as the main plant species was related to its inherent plant characteristics. In the lower elevation riparian zone, where the duration of flooding was longer and species diversity was lower, *C. dactylon* exhibited strong tolerance to flooding. Even when the riparian zone soil remained inundated for more than 200 days during winter, *C. dactylon* could still grow during the following summer when water was released from the reservoir. However, when the elevation exceeded 165 m, the duration of soil inundation became shorter, and the number of viable species increased, leading to greater interspecific competition and the loss of dominance by *C. dactylon*. *Hemarthria compressa*, *S. viridis*, and *Euphorbia humifusa* became the dominant plant species at elevations of 170 m or above.

From 145 m to 180 m elevations, the length of the upright stems of *C. dactylon* increased with elevation (yellow loam riparian zone: R$^2$ = 0.494, *p* < 0.05; purple soil riparian zone: R$^2$ = 0.674, *p* < 0.05). The number of tillers of *C. dactylon* varied significantly across different elevations. In the yellow loam riparian zone, the number of tillers initially decreased and then increased with elevation (R$^2$ = 0.664, *p* < 0.05). In the purple soil riparian zone, the

number of tillers significantly decreased with increasing elevation ($R^2$ = 0.944, $p$ < 0.001). The average internode length showed a trend of increasing with elevation (yellow loam riparian zone: $R^2$ = 0.262, $p$ = 0.137; purple soil riparian zone: $R^2$ = 0.460, $p$ = 0.084). From the 145 m to 175 m elevations, the height of *C. dactylon* increased with elevation (yellow loam riparian zone: $R^2$ = 0.314, $p$ = 0.111; purple soil riparian zone: $R^2$ = 0.891, $p$ < 0.01). The primary stolon length of *C. dactylon* exhibited a trend of decreasing followed by increasing with elevation (yellow loam riparian zone: $R^2$ = 0.463, $p$ < 0.05; purple soil riparian zone: $R^2$ = 0.714, $p$ < 0.05). The root length of *C. dactylon* showed a trend of increasing with elevation in both the yellow loam riparian zone ($R^2$ = 0.370, $p$ = 0.087) and the purple soil riparian zone ($R^2$ = 0.648, $p$ < 0.05). Figure 5 illustrates the variations in the morphological characteristics of *C. dactylon* under different soil types and elevations of water level.

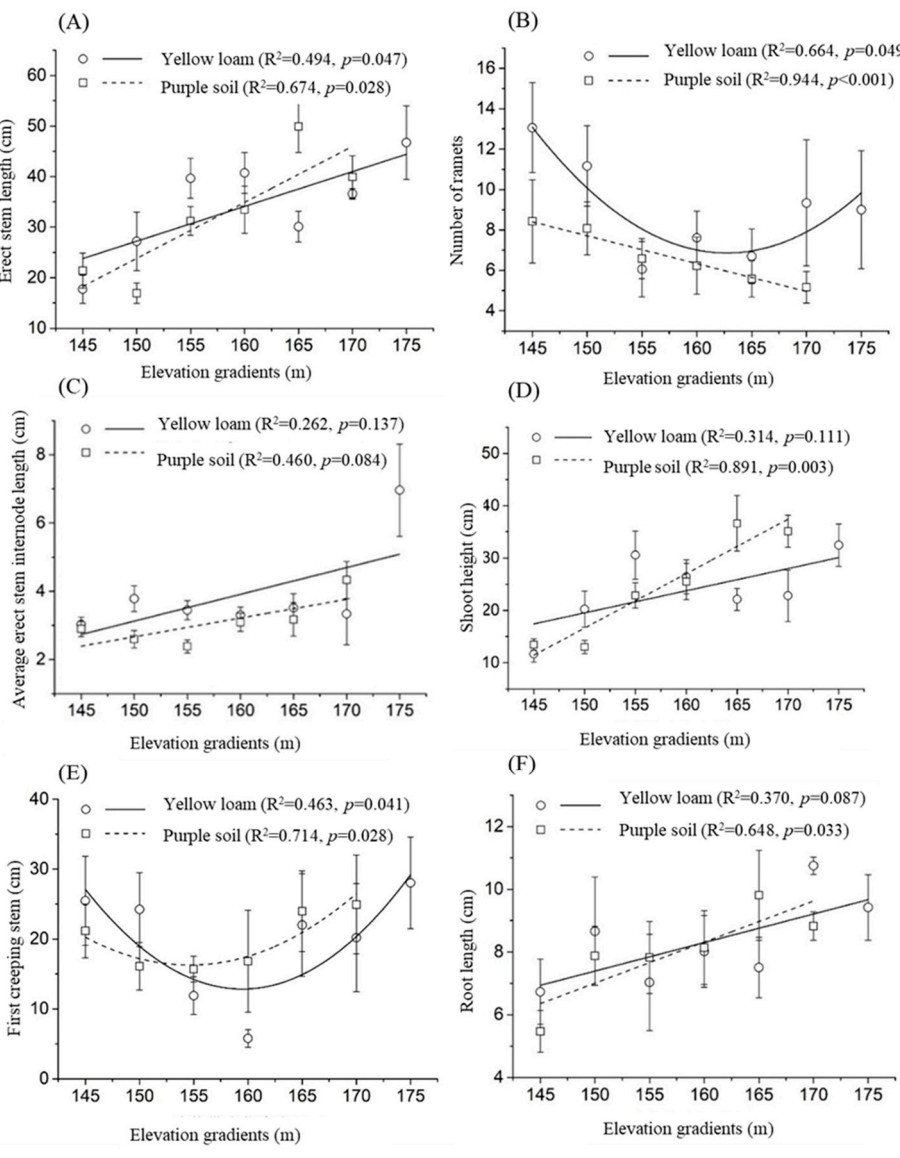

**Figure 5.** Variation patterns of *C. dactylon* morphology at different elevations of water level. Data are shown as means ± SE (N = 54). (**A**) Erect stem length, (**B**) number of ramets, (**C**) average erect stem internode length, (**D**) shoot height, (**E**) first creeping stem, and (**F**) root length of *C. dactylon* at different elevations of water level.

*3.4. Relationship between Riparian Plant Biomass and C. dactylon Morphological Traits with Environmental Factors*

3.4.1. Relationship between Riparian Plant Biomass and Environmental Factors

The relationships between plant biomass and environmental factors are shown in Figure 6. The biomass of plant *C. dactylon* in the Daning River was significantly positively correlated with soil total nitrogen content (R = 0.44, $p < 0.05$) and flood duration (R = 0.4, $p < 0.05$). Plant biomass also showed a positive correlation with soil moisture content and temperature. The biomass of plant *C. dactylon* was negatively correlated with soil pH, electrical conductivity, and total phosphorus.

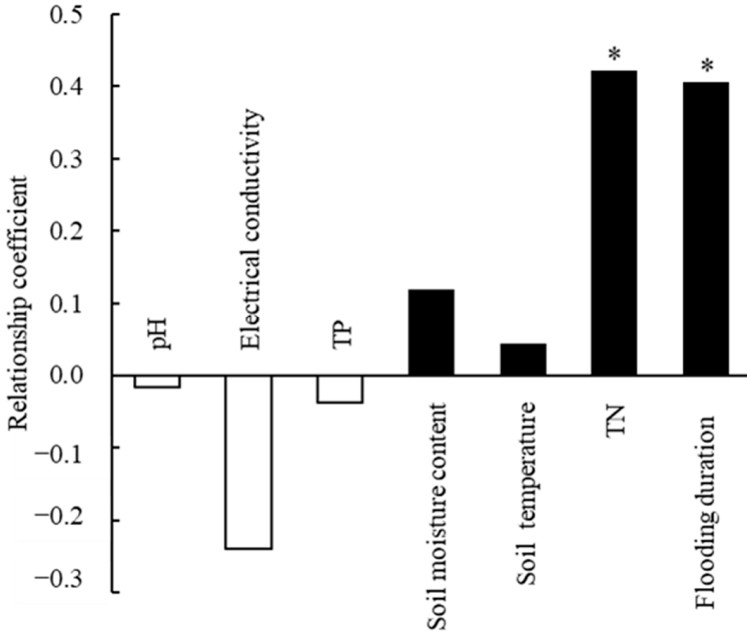

**Figure 6.** Relationship between plant biomass distribution and environmental factors in the riparian zone of the Daning River. Data are shown as means $\pm$ SE ($N = 36$). * $p < 0.05$.

3.4.2. Relationship between *C. dactylon* Morphological Traits and Environmental Factors

The Spearman's rank correlation between the morphological characteristics of *C. dactylon* and soil environmental factors is shown in Figure 7. Results indicated a correlation between the morphological traits of *C. dactylon* and environmental factors. The plant height of *C. dactylon* was highly positively correlated with its upright stem length (R = 0.95, $p < 0.001$) and significantly negatively correlated with its first creeping stem length (R = −0.44, $p < 0.05$). The upright stem length (R = −0.48, $p < 0.05$) and plant height (R = −0.47, $p < 0.05$) of *C. dactylon* were significantly negatively correlated with soil moisture content.

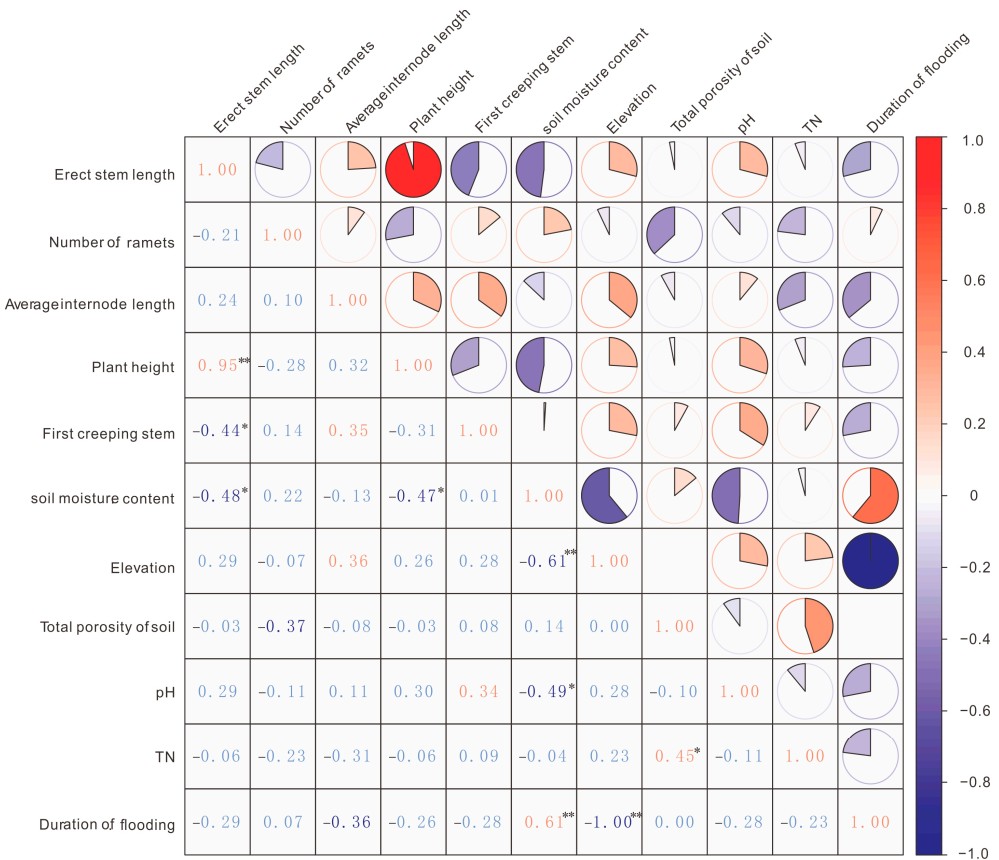

**Figure 7.** Spearman's rank correlation between *C. dactylon* morphological traits and soil environmental factors. $N = 108$, ** $p < 0.01$; * $p < 0.05$.

## 4. Discussion

### 4.1. Influence of Reservoir Water Level Rhythm on the Survival and Growth of Plants in the Riparian Zones

The periodic fluctuation of water levels in the Three Gorges Reservoir disrupts the natural flood–drought pattern of rivers and creates a specific reservoir water level rhythm. The impact of water level elevation on plant communities may be related to resource differentiation and vegetation ecological adaptation differences [1]. In the lower part of the riparian zone, where flooding stress was intense, the establishment of vegetation was hindered, and intolerant plant species perished due to the lack of organismal structures and functions that adapted to extreme environments, resulting in simplified community composition. The upper part of the riparian zone, where microhabitat conditions were more complex and resource combinations were optimal, was conducive to the establishment and growth of vegetation species with a wider ecological niche, leading to a higher species diversity in the community [19,20]. The results of this study indicated that there was a correlation among the main influencing factors (duration of flooding, elevation of water level, and soil moisture content) that affected the spatial distribution of plant communities, and they are all related to the hydrological characteristics of the reservoir. As the elevation of the water level increased, flooding duration, frequency, and depth decreased (Figure 3). Soil moisture content also showed a strong correlation with hydrological factors such as flooding duration, frequency, and depth. For example, soil moisture content was significantly negatively correlated with water level elevation ($R = -0.61$, $p < 0.001$) and significantly positively correlated with the number of days of inundation ($R = 0.61$, $p < 0.001$). These results are consistent with previous studies. Capon [21] and Su et al. [22] considered the duration of flooding as the main factor influencing plant community composition and diversity. In their study on the species richness patterns of riparian plant communities

in the Pengxi River, Tong et al. [23] found that flooding duration, soil moisture content, and substrate heterogeneity had important effects on the distribution patterns of plant communities. Wang and Hong [24] found in their study on the effects of the Three Gorges Dam on vegetation coverage at different elevations in the riparian zone that an increase in water level had a negative impact on vegetation coverage below an elevation of 175 m.

*4.2. Impact of Multiple Environmental Stressors on the Vegetation Biomass in the Riparian Zone*

Previous studies have shown various forms of relationship between riparian plant biomass and elevation of water level [25,26]. It is generally believed that they exhibit a negative correlation, meaning that species diversity decreases with increasing water level [27]. Another form is the "mid-domain bulge", where biomass initially increases and then decreases with increasing water level [28]. The conclusions drawn in this study align with the "mid-domain bulge" theory. Influenced by the fluctuation of water levels in the Three Gorges Reservoir (Figure 4), the diversity index of vegetation in the riparian zone reached its lowest value in the low elevation zone (145~155 m). After experiencing long-term annual water level fluctuations, vegetation biomass in the reservoir riparian zone showed a trend of initially increasing and then decreasing with increasing elevation, reaching its highest point in the mid-elevation zone (155~165 m).

The seasonal inundation–exposure regime in the riparian zone of the Three Gorges Reservoir created specific macro-habitats. However, soil erosion, sediment deposition, and changes in the soil matrix environment have increased habitat fragmentation and vulnerability, making the ecosystem more fragile and sensitive [29]. The periodic rise and fall of water levels disrupted the stability of soil structure through water erosion and sediment deposition, leading to soil nutrient loss and unstable growth substrates [30,31]. The accumulated duration of flooding formed during the periodic rise and fall of water levels (Figure 3) primarily affects soil physical properties such as soil moisture content and porosity. For example, with the increasing duration of flooding, soil moisture content showed a significant positive correlation ($R = 0.61$, $p < 0.01$). It further affected soil chemical properties, such as organic matter and nutrient content. Soil moisture content was significantly negatively correlated with soil pH ($R = -0.49$, $p < 0.05$), and soil bulk density was significantly negatively correlated with total nitrogen content ($R = -0.45$, $p < 0.05$). Therefore, the accumulated duration of flooding and average flooding depth formed by the periodic rise and fall of water levels were the primary stressors determining the differentiation of vegetation biomass along the elevation gradient. The intense and prolonged flooding and delayed exposure in the lower part of the riparian zone hindered the photosynthesis and metabolic processes of vegetation. To reduce energy consumption, plants adopted strategies such as reducing population density and allocating more resources to reproduction, thereby slowing down plant growth [30]. In the upper part of the riparian zone, vegetation was greatly affected by land-based infrastructure and human factors, and the low moisture content was unfavorable for plant growth and nutrient uptake, resulting in a distribution trend of low biomass at both ends and high biomass in the middle.

Soil, as an important component of material and energy cycling in the riparian zone, plays a crucial role in coordinating plant growth and supplying nutrients, thus determining the productivity of plant communities. The periodic rise and fall of water levels lead to the breakdown of large-sized soil aggregates into microaggregates, accelerating the release, transport, and diffusion of soil nutrients and resulting in nutrient-poor soil conditions in the riparian zone. The positive correlation between vegetation biomass and duration of flooding and total nitrogen content indicated that these factors were the main soil limiting factors determining the differentiation of vegetation biomass along the elevation gradient. High concentrations of total nitrogen can stimulate seed germination [32], promote root absorption to maintain plant nitrogen stoichiometry balance, and enhance chlorophyll synthesis to increase ecosystem productivity. Thus, soil total nitrogen is in line with changes in vegetation biomass. On the other hand, the duration of flooding can increase soil moisture content. As an important carrier of energy cycling, soil moisture content affects

the transformation and transport of nutrients in the soil, thereby determining the efficiency of vegetation in utilizing soil water and nutrients and promoting biomass accumulation.

*4.3. Adaptation Mechanisms of C. dactylon Morphology in the Reservoir Riparian Zones*

The morphological mechanisms by which plants in the riparian zone adapt to flooding stress involve various processes that facilitate gas transport to avoid hypoxia. These mechanisms include the development of adventitious roots, the formation of root and leaf aerenchyma, and the formation of leaf gas films, all of which enhance oxygen and carbon dioxide exchange in plants and maintain root aeration status [33]. During the flooding period, the root aerenchyma tissue of *C. dactylon* develops [34] and root biomass increases [35], indicating that the roots remain vital, enabling quick sprouting after the riparian zone is exposed. Additionally, during flooding, the aboveground stolons of *C. dactylon* quickly die off, while the underground rhizomes firmly anchor in the soil, absorbing soil nutrients and storing energy. When the aboveground stems of *C. dactylon* emerge from the water surface, photosynthesis and aerobic respiration can take place, facilitating rapid plant growth.

Plant height, upright stem length, primary stolon length, and root length of *C. dactylon* in the riparian zone were positively correlated with elevation and negatively correlated with flooding duration. The dominance of *C. dactylon* in the 145~155 m elevation of the Three Gorges Reservoir riparian zone may be related to its internal aerenchyma tissue [36–38]. In the low elevation zone, the plant height and average internode length of *C. dactylon* decreased, whereas the number of tillers increased. This facilitated the rapid diffusion of gases from stems to roots, providing oxygen for root respiration and promoting root growth. Consequently, it increased tissue porosity and oxygen leakage, enhancing the porosity of primary and adventitious roots and the waterlogging tolerance of the plant. In the high elevation zone, as the duration of waterlogging decreased, the time available for photosynthesis and aerobic respiration increased, leading to more pronounced growth. Considering the growth pattern of *C. dactylon* under flooding conditions, shorter flooding durations allowed for a longer period of photosynthesis in the aboveground stems, resulting in longer vegetation growth. This aligns with the findings of this study.

## 5. Conclusions

This study focused on the riparian zone of the Daning River, a typical tributary in the Three Gorges Reservoir area. Through field investigations and laboratory analysis, the distribution characteristics of riparian vegetation under different soil types and elevations of water level were revealed. *C. dactylon*, as the dominant species, was selected for studying its plant characteristics under different soil types and elevations, and the environmental factors influencing its growth were explored. The main conclusions are as follows:

(1) There was no significant difference in plant biomass and morphological indicators between the yellow loam and purple soil riparian zones along the same elevation gradient.

(2) Plant biomass showed a trend of initial increase followed by a decrease with increasing elevation of water level, reaching a higher value in the riparian zone at an elevation of 155–165 m.

(3) Soil total nitrogen content was identified as a key limiting factor for plant biomass.

(4) The plant height and stolon length of *C. dactylon* decreased with increasing flooding duration, facilitating the rapid diffusion of gases from stems to roots, providing oxygen for root respiration, and adapting to long-term flooding in the riparian zone. Consequently, *C. dactylon* became the dominant species in the 145–155 m elevation range in the riparian zone.

**Author Contributions:** Conceptualization, X.L., S.L. and Z.L.; methodology, X.L. and S.L.; software, X.L., S.L. and Z.W.; validation, Y.X. and Z.L.; formal analysis, X.L., S.L. and Z.W.; investigation, X.L. and S.L.; resources, S.L., Z.L. and Y.X.; data curation, X.L. and S.L.; writing—original draft preparation, X.L., S.L. and Z.W.; writing—review and editing, X.L., S.L., Y.X., Z.W. and Z.L.; visualization, X.L., S.L. and Z.W.; supervision, Z.L.; project administration, S.L. and Z.L.; funding acquisition, S.L., Y.X. and Z.L. All authors have read and agreed to the published version of the manuscript.

**Funding:** This research was funded by the National Natural Science Foundation of China project (No. 51809287, No. 42202269), the Young Elite Scientists Sponsorship Program by CAST (No. 2022QNRC001), and the Follow-up Work of the Three Gorges Project (No. 2136902).

**Data Availability Statement:** The data that support the findings of this study are available from the corresponding author upon reasonable request.

**Acknowledgments:** We thank Aimin Cai, Yijie Liu and Pengcheng Du for their help with sample collection. We also thank Xiaoru Su for the English translation.

**Conflicts of Interest:** The authors declare no conflict of interest.

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
