# Peer review of "What Drives the Morphological Traits of Stress-Tolerant Plant Cynodon dactylon in a Riparian Zone of the Three Gorges Reservoir, China"

_water, doi:10.3390/w15183183_

Round 1

Reviewer 1 Report

Very interesting and valuable paper. Very well chosen methodology and very sophisticated working out of obtained results. The results of the evaluated study can really represent scientific support and inspiration not only for the ecological protection of the environment of the fluctuation zone in the Three Gorges Dam, but also in other similar cases.

Author Response

Thank you very much for your positive comments. We have improved the quality of the manuscript to meet the Journal Water requirement.

Reviewer 2 Report

I ve done my cooments in the text

Author Response

Reviewer #2: I’ve done my cooments in the text.

Response: Thank you very much for your careful comments. We have revised our manuscript according to your comments. See below for more details.

Italic the scientific name and if its mention for the first time in the paper put the author of the species-(L.) Pers.

Response: Thank you very much for your careful comments. We have revised the plant scientific name for the first time in the paper as “Cynodon dactylon (L) Pers”. And italic the other plant scientific name in the whole paper short as “C. dactylon”.

Add apace-bar between “60 ℃” and “(Li et al. 2016)”.

Response: Added.

2.3 Data analysis and processing

Delete “In addition”

Response: Deleted.

The plant biomass in Yellow loam peaks at an elevation of 160 m, approximately 1395.40 g/cm2, while the biomass in Purple soil peaks at an elevation of 155 m, approximately 1291.17 g/cm2. Superscript no. 2.

Response: We have revised it as “The plant biomass in yellow loam peaked at an elevation of 160 m to approximately 1,395.40 g/cm2, whereas the biomass in purple soil peaked at an elevation of 155 m to approximately 1,291.17 g/cm2.”

Italic and mention the author who describe the species first time.

Response: We have revised it as “”After the water impoundment operation of the Three Gorges Reservoir in 2003, the riparian zone gradually formed a vegetation succession pattern dominated by a few species including the plant Cynodon dactylon (L) Pers, accompanied by Atractylodes macrocephala Koidz. and Homalomena aromatica Gagnep”“ At elevations of 170 m and above, higher plant diversity was observed, including species such as Daucus carota L., Ambrosia artemisiifolia L., Conyza canadensis (L.) Cronq., Digitaria sanguinalis (L.) Scop., Bidens frondosa L., Vitex negundo L., Melilotus officinalis (L.) Desr., Beckmannia syzigachne (Steud.) Fern., and Setaria viridis (L.) Beauv.”

Figure 5 in text should be mention before the actual figure.

Response: We have deleted the sentence “The morphological characteristics of Cynodon dactylon under different soil types and ele-vations are shown in Fig.5.” and revised the last sentence of this paragraph as “Figure 5 illustrates the variations in the morphological characteristics of C. dactylon under different soil types and water level elevations.” And we moved the actual figure 5 under the text.

Delete the “(Figure 6)”in the sentence of discussion 4.2.

Response: We have deleted the “(Figure 6)” in the sentence of discussion 4.2.

Reviewer 3 Report

I have reviewed manuscript water-2546935 by Li et al. The manuscript concerns the annual cycle of roughly 30-meter changes in the depth of water behind the Three Gorges Dam and the effect of those changes on the plant community in the sediments/soil of the “fluctuation zone”. The authors find that the biomass of vegetation is highest in the mid-elevation zone, and they discuss the adaptations of Cynodon dactylon to the cycle of immersion and exposure to air.  Because the study was carried out over a period of three years, 2020–2023,  it was possible to confirm the reproducibility of the phenomena. I think the science is sound. I did find myself asking, so what? The last sentence of the Introduction suggests an answer: “The findings of this study are expected to provide scientific  support for the ecological environment protection of the fluctuation zone in the Three  Gorges Reservoir.” I am unsure what this is supposed to imply, and I think that is the weakness of this manuscript. One of the purposes of the Three Gorges Dam is flood control, and my impression is that the principal reason for the controlled variations of the water level behind the dam is flood control. What are the authors hoping will happen in the way of “ecological environment protection” that is not happening now? I think some clarification of that point would strengthen the manuscript.

Author Response

Response: Thank you very much for your comments. We now have revised the whole manuscript and make it clearer and easier to understand for readers. We rewrote the Introduction section as follows:

The riparian zone of a reservoir, also known as the drawdown zone or disturbance zone, is a transitional ecosystem that alternately experiences submergence and exposure due to periodic fluctuations in water level [1]. These fluctuations can result from natural hydrological changes, such as seasonal water level fluctuations, or anthropogenic manipulations, primarily due to cyclical water storage and discharge operations. Additionally, specific climatic events such as droughts can cause drawdown in reservoirs.

The Three Gorges Project is a key national infrastructure and a critical node in the water network of China. The Three Gorges Reservoir serves as an important strategic freshwater resource reserve for the nation and is a crucial ecological barrier in the upper reaches of the Yangtze River [2]. In 2010, the project achieved its experimental water storage target of 175 meters for the first time. Since then, to maximize flood control benefits, the Three Gorges Reservoir has adopted a “winter storage and summer discharge” regulatory approach. The reservoir operates at a lower water level from April to October and at a higher water level from October to the following April. This unique operational scheduling forms a unique fluctuating environment in the riparian zone with a drop range of 30 m, between 145 m and 175 m [1].

The cyclic water impoundment and release during normal operation of the Three Gorges Reservoir creates a unique form of habitat stress, altering the structure and function of the ecological pattern of vegetation in the riparian zone. In the cross-section of the riparian zone at different elevations, the ecological pattern of the vegetation exhibits notable spatial heterogeneity [3]. The biodiversity in the riparian zone is significantly affected by hydrological disturbances. Large-scale, unnatural water level fluctuations severely disrupt the original distribution of plant communities, resulting in a reverse succession in the riparian zone [4-6]. After the water impoundment operation of the Three Gorges Reservoir in 2003, the riparian zone gradually formed a vegetation succession pattern dominated by a few species including the plant Cynodon dactylon (L) Pers, accompanied by Atractylodes macrocephala Koidz. and Homalomena aromatica Gagnep [1,7]. Since C. dactylon is the most dominant plant species, covering almost all the riparian zones of the Three Gorges Reservoir, we selected it as our study object. The plant C. dactylon shows significant resistance to flooding and nutrient-poor conditions [8,9]. This might have important effects on the material conversion and ecological effects of element cycling in the riparian zone [10]. During the flooding period, the submerged plants will release several nutrients into the water, resulting in water eutrophication [11]. During the water-level drawdown period, plants are in their growing season.

The growth behavior and life history strategies of plants at each stage of their growth and development are closely related to the morphological traits of plants and can determine the distribution pattern and population behavior of plant species in the habitat to a certain extent. The plant life history strategy is the best resource allocation method for plant species to maintain growth and reproduction. The adaptability of species is accumulated in the process of species evolution, which is manifested in morphology. Different morphological growth strategies are adopted by plants to maximize the fitness of species at certain stages. Plant morphological traits play an important role in studying biogeography in many ecosystems such as forestry [12] and agricultural [13] ecosystems. Under drought stress, plants can obtain the water content required for normal growth and development through the plasticity of morphological and structural characteristics, and adapt to the stressed environment by changing their morphogenesis. However, the morphological traits of plants in the riparian zone of reservoirs have received little attention. Differences in abiotic factors caused by climate change and geographical location differences will lead to different morphological characteristics of vegetation [14, 15].

To maintain the C. dactylon community, it is important to address the morphological traits of this plant across a gradient of elevations, as well as its relationships with regional abiotic factors in the riparian zones. Hence, in this study, we tested the hypotheses that (1) C. dactylon morphological traits varied across elevations between two types of soil (yellow loam and purple soil) and (2) C. dactylon morphological traits varied with abiotic drivers.

Reviewer 4 Report

In the manuscript “ Adaptive characteristics of stress-tolerant plants in the water- level fluctuation zone of Three Gorges Reservoir”, authors Xiaolong Li , Shanze Li, Yawei Xie, Zehui Wei and Zilong Li, discussed the factors affecting the growth of plants in the fluctuation zone and the adaptive mechanisms of Bermuda grass to the reservoir fluctuation zone.

Abstract

L 16 The following sentence is unclear: Results indicate that dominant plants varied with elevation.

Please, write scientific names in Latin!

Introduction

Clear and concise.

Materials and methods

The numbers next to Figure 1 are reversed

In Figure 2, there are Chinese letters.

L 104 … at 60 ℃(Li et al. 2016). Write spce before “(“

Results

Captions are missing for figures (what different letters mean, what was compared, what is N,…)

Are Atractylodes macrocephala and Homalomena aromatica grasses? Please, clarify that!

L 199 loss of dominance by Cynodon dactylon.. Delete one full stop. And which species then dominate?

Certain things are repeated in the results chapter.

L 229 The biomass of riparian plants in the Daning … Please, write which plants!

The results are not clearly presented.

Discussion

Based on the results, the discussion is insufficiently in-depth. Many sentences from the Discussion chapter fit better in the Introduction chapter.

Specific comments

The article addresses an interesting topic with practical significance, bringing strengths to the study. The article has an insufficient discussion about other plants that grow in the studied area. There is no clear concept in the article.

My suggestion: major revision

Some statements are not clear.

Author Response

L 16 The following sentence is unclear: Results indicate that dominant plants varied with elevation.

Response: We revised the Abtract section as “The cyclical process of water storage and recession in the regular operation of the Three Gorges Reservoir creates a unique habitat stress that alters the structural and functional attributes of vegetation ecology within the riparian zone. The stress-tolerant plant Cynodon dactylon (L) Pers is the dominant plant species in the riparian zone of the Three Gorges Reservoir. In this study, the riparian zone of the Daning River, a tributary located central to the Three Gorges Reservoir, was selected as our study area. To identify the drivers of the morphological traits of C. dactylon in the riparian zone of Daning River, we examined plant biomass and plant characteristics across different elevation gradients, with reference to abiotic factors to determine the distribution patterns of plant morphological traits. Results indicated that in the two main soil types of the riparian zone, plant biomass showed a consistent trend along the elevation gradient, with a “middle-height expansion” pattern; biomass increased and then decreased with rising water levels. Plant biomass positively correlated with soil total nitrogen and negatively correlated with soil pH, electrical conductivity, and total phosphorus. C. dactylon adapted to prolonged flooding in the riparian zone by having a significant negative correlation between plant height and erect stem length with soil moisture content to facilitate root respiration.”

Please, write scientific names in Latin!

Response: We also wrote the scientific name in Latin as Cynodon dactylon (L) Pers.

Introduction

Clear and concise.

Response: Thank you very much for your comments. We have revised the Introduction section and made it clear and concise. Detailed as follows:

The riparian zone of a reservoir, also known as the drawdown zone or disturbance zone, is a transitional ecosystem that alternately experiences submergence and exposure due to periodic fluctuations in water level [1]. These fluctuations can result from natural hydrological changes, such as seasonal water level fluctuations, or anthropogenic manipulations, primarily due to cyclical water storage and discharge operations. Additionally, specific climatic events such as droughts can cause drawdown in reservoirs.

The Three Gorges Project is a key national infrastructure and a critical node in the water network of China. The Three Gorges Reservoir serves as an important strategic freshwater resource reserve for the nation and is a crucial ecological barrier in the upper reaches of the Yangtze River [2]. In 2010, the project achieved its experimental water storage target of 175 meters for the first time. Since then, to maximize flood control benefits, the Three Gorges Reservoir has adopted a “winter storage and summer discharge” regulatory approach. The reservoir operates at a lower water level from April to October and at a higher water level from October to the following April. This unique operational scheduling forms a unique fluctuating environment in the riparian zone with a drop range of 30 m, between 145 m and 175 m [1].

The cyclic water impoundment and release during normal operation of the Three Gorges Reservoir creates a unique form of habitat stress, altering the structure and function of the ecological pattern of vegetation in the riparian zone. In the cross-section of the riparian zone at different elevations, the ecological pattern of the vegetation exhibits notable spatial heterogeneity [3]. The biodiversity in the riparian zone is significantly affected by hydrological disturbances. Large-scale, unnatural water level fluctuations severely disrupt the original distribution of plant communities, resulting in a reverse succession in the riparian zone [4-6]. After the water impoundment operation of the Three Gorges Reservoir in 2003, the riparian zone gradually formed a vegetation succession pattern dominated by a few species including the plant Cynodon dactylon (L) Pers, accompanied by Atractylodes macrocephala Koidz. and Homalomena aromatica Gagnep [1,7]. Since C. dactylon is the most dominant plant species, covering almost all the riparian zones of the Three Gorges Reservoir, we selected it as our study object. The plant C. dactylon shows significant resistance to flooding and nutrient-poor conditions [8,9]. This might have important effects on the material conversion and ecological effects of element cycling in the riparian zone [10]. During the flooding period, the submerged plants will release several nutrients into the water, resulting in water eutrophication [11]. During the water-level drawdown period, plants are in their growing season.

The growth behavior and life history strategies of plants at each stage of their growth and development are closely related to the morphological traits of plants and can determine the distribution pattern and population behavior of plant species in the habitat to a certain extent. The plant life history strategy is the best resource allocation method for plant species to maintain growth and reproduction. The adaptability of species is accumulated in the process of species evolution, which is manifested in morphology. Different morphological growth strategies are adopted by plants to maximize the fitness of species at certain stages. Plant morphological traits play an important role in studying biogeography in many ecosystems such as forestry [12] and agricultural [13] ecosystems. Under drought stress, plants can obtain the water content required for normal growth and development through the plasticity of morphological and structural characteristics, and adapt to the stressed environment by changing their morphogenesis. However, the morphological traits of plants in the riparian zone of reservoirs have received little attention. Differences in abiotic factors caused by climate change and geographical location differences will lead to different morphological characteristics of vegetation [14, 15].

To maintain the C. dactylon community, it is important to address the morphological traits of this plant across a gradient of elevations, as well as its relationships with regional abiotic factors in the riparian zones. Hence, in this study, we tested the hypotheses that (1) C. dactylon morphological traits varied across elevations between two types of soil (yellow loam and purple soil) and (2) C. dactylon morphological traits varied with abiotic drivers.

Materials and methods

The numbers next to Figure 1 are reversed

Response: Thank you very much for your comments. Even though I am not quite sure what do you mean, we updated the Figure 1.

In Figure 2, there are Chinese letters.

Response: Thank you very much for your comments. We updated the Figure 2.

L 104 … at 60 ℃(Li et al. 2016). Write spce before “(“

Response: Thank you very much for your careful comments. We added a spce before “(”.

Results

Captions are missing for figures (what different letters mean, what was compared, what is N,…)

Response: Thank you very much for your careful comments. We have revised all the captions of figures. For figure 4. We added “Data are shown as means ± SE (n = 54). All ANOVA tests were significant (P < 0.05 in each case). The letter above each bar represents the results of post hoc Tukey’s HSD test: bars sharing a letter are not significantly different from one another.” For figure 5, we added “Data are shown as means ± SE (n = 54).” For figure 6, we added “Data are shown as means ± SE (n = 36). *: P<0.05.” For figure 7, we added “N=108”.

Are Atractylodes macrocephala and Homalomena aromatica grasses? Please, clarify that!

Response: Thank you very much for your careful comments. We have revised the paragraph as “After the water impoundment operation of the Three Gorges Reservoir in 2003, the riparian zone gradually formed a vegetation succession pattern dominated by a few species including the plant Cynodon dactylon (L) Pers, accompanied by grasses Atractylodes macrocephala Koidz. and Homalomena aromatica Gagnep”

L 199 loss of dominance by Cynodon dactylon.. Delete one full stop. And which species then dominate?

Response: Thank you very much for your careful comments. We deleted the “.”.

We revised the paragraph as “The investigation results revealed that the dominant plant species within the 145 m to 165 m elevation range in the riparian zone of the Daning River were primarily C. dactylon, occasionally accompanied by A. macrocephala and H. aromatica. At elevations of 170 m and above, higher plant diversity was observed, including species such as Daucus carota L., Ambrosia artemisiifolia L., Conyza canadensis (L.) Cronq., Digitaria sanguinalis (L.) Scop., Bidens frondosa L., Vitex negundo L., Melilotus officinalis (L.) Desr., Beckmannia syzigachne (Steud.) Fern., and Setaria viridis (L.) Beauv. These species are mainly annual and perennial herbaceous plants. At higher elevations, shrubs, trees, and farmland were also present. ”

Certain things are repeated in the results chapter.

Response: Thank you very much for your careful comments. We deleted the sentence “This study focuses on C. dactylon as the primary species and investigates its plant characteristics under different soil types and elevations, aiming to explore the environmental factors influencing its growth.”

L 229 The biomass of riparian plants in the Daning … Please, write which plants!

Response: Thank you very much for your careful comments. We rewrote the sentence as “The biomass of plant C. dactylon in the Daning River was significantly positively correlated with soil total nitrogen content (R = 0.44, P < 0.05) and flood duration (R = 0.4, P < 0.05).” See new 3.4.1 section.

The results are not clearly presented.

Response: Thank you very much for your comments. We have revised the results section to make it clear. See new Results section.

Discussion

Based on the results, the discussion is insufficiently in-depth. Many sentences from the Discussion chapter fit better in the Introduction chapter.

Specific comments

The article addresses an interesting topic with practical significance, bringing strengths to the study. The article has an insufficient discussion about other plants that grow in the studied area. There is no clear concept in the article.

Response: Thank you very much for your comments. We have revised the discussion section to make it clear. See new discussion section.

  1. Discussion

4.1. Influence of reservoir water level rhythm on the survival and growth of plants in the riparian zones

The periodic fluctuation of water levels in the Three Gorges Reservoir disrupts the natural flood-drought pattern of rivers and creates a specific reservoir water level rhythm. The impact of water level elevation on plant communities may be related to resource differentiation and vegetation ecological adaptation differences [1]. In the lower part of the riparian zone, where flooding stress was intense, the establishment of vegetation was hindered, and intolerant plant species perished due to the lack of organismal structures and functions that adapted to extreme environments, resulting in simplified community composition. The upper part of the riparian zone, where microhabitat conditions were more complex and resource combinations were optimal, was conducive to the establishment and growth of vegetation species with a wider ecological niche, leading to a higher species diversity in the community[19,20]. The results of this study indicated that there was a correlation among the main influencing factors (duration of flooding, elevation of water level, and soil moisture content) that affected the spatial distribution of plant communities, and they are all related to the hydrological characteristics of the reservoir. As the elevation of the water level increased, flooding duration, frequency, and depth decreased (Figure 3). Soil moisture content also showed a strong correlation with hydrological factors such as flooding duration, frequency, and depth. For example, soil moisture content was significantly negatively correlated with water level elevation (R = -0.61, P < 0.001) and significantly positively correlated with the number of days of inundation (R = 0.61, P < 0.001). These results are consistent with previous studies. Capon [21] and Su et al. [22] considered the duration of flooding as the main factor influencing plant community composition and diversity. In their study on the species richness patterns of riparian plant communities in the Pengxi River, Tong et al.[23] found that flooding duration, soil moisture content, and substrate heterogeneity had important effects on the distribution patterns of plant communities. Wang and Hong [24] found in their study on the effects of the Three Gorges Dam on vegetation coverage at different elevations in the riparian zone that an increase in water level had a negative impact on vegetation coverage below an elevation of 175 m.

4.2. Impact of multiple environmental stressors on the vegetation biomass in the riparian zone

Previous studies have shown various forms of relationship between riparian plant biomass and elevation of water level [25, 26]. It is generally believed that they exhibit a negative correlation, meaning that species diversity decreases with increasing water level [27]. Another form is the “mid-domain bulge,” where biomass initially increases and then decreases with increasing water level [28]. The conclusions drawn in this study align with the “mid-domain bulge” theory. Influenced by the fluctuation of water levels in the Three Gorges Reservoir (Figure 4), the diversity index of vegetation in the riparian zone reached its lowest value in the low elevation zone (145~155 m). After experiencing long-term annual water level fluctuations, vegetation biomass in the reservoir riparian zone showed a trend of initially increasing and then decreasing with increasing elevation, reaching its highest point in the mid-elevation zone (155~165 m).

The seasonal inundation-exposure regime in the riparian zone of the Three Gorges Reservoir created specific macro-habitats. However, soil erosion, sediment deposition, and changes in the soil matrix environment have increased habitat fragmentation and vulnerability, making the ecosystem more fragile and sensitive [29]. The periodic rise and fall of water levels disrupted the stability of soil structure through water erosion and sediment deposition, leading to soil nutrient loss and unstable growth substrates [30, 31]. The accumulated duration of flooding formed during the periodic rise and fall of water levels (Figure 3) primarily affecting soil physical properties such as soil moisture content and porosity. For example, with the increasing duration of flooding, soil moisture content showed a significant positive correlation (R = 0.61, P < 0.01). It further affected soil chemical properties, such as organic matter and nutrient content. Soil moisture content was significantly negatively correlated with soil pH (R = -0.49, P < 0.05), and soil bulk density was significantly negatively correlated with total nitrogen content (R = -0.45, P < 0.05). Therefore, the accumulated duration of flooding and average flooding depth formed by the periodic rise and fall of water levels were the primary stressors determining the differentiation of vegetation biomass along the elevation gradient. The intense and prolonged flooding and delayed exposure in the lower part of the riparian zone hindered the photosynthesis and metabolic processes of vegetation. To reduce energy consumption, plants adopted strategies such as reducing population density and allocating more resources to reproduction, thereby slowing down plant growth [30]. In the upper part of the riparian zone, vegetation was greatly affected by land-based infrastructure and human factors, and the low moisture content was unfavorable for plant growth and nutrient uptake, resulting in a distribution trend of low biomass at both ends and high biomass in the middle.

Soil, as an important component of material and energy cycling in the riparian zone, plays a crucial role in coordinating plant growth and supplying nutrients, thus determining the productivity of plant communities. The periodic rise and fall of water levels lead to the breakdown of large-sized soil aggregates into microaggregates, accelerating the release, transport, and diffusion of soil nutrients and resulting in nutrient-poor soil conditions in the riparian zone. The positive correlation between vegetation biomass and duration of flooding and total nitrogen content indicated that these factors were the main soil limiting factors determining the differentiation of vegetation biomass along the elevation gradient. High concentrations of total nitrogen can stimulate seed germination [32], promote root absorption to maintain plant nitrogen stoichiometry balance, and enhance chlorophyll synthesis to increase ecosystem productivity. Thus, soil total nitrogen is in line with changes in vegetation biomass. On the other hand, the duration of flooding can increase soil moisture content. As an important carrier of energy cycling, soil moisture content affects the transformation and transport of nutrients in the soil, thereby determining the efficiency of vegetation in utilizing soil water and nutrients and promoting biomass accumulation.

4.3. Adaptation mechanisms of C. dactylon morphology in the reservoir riparian zones

The morphological mechanisms by which plants in the riparian zone adapt to flooding stress involve various processes that facilitate gas transport to avoid hypoxia. These mechanisms include the development of adventitious roots, the formation of root and leaf aerenchyma, and the formation of leaf gas films, all of which enhance oxygen and carbon dioxide exchange in plants and maintain root aeration status [33]. During the flooding period, the root aerenchyma tissue of C. dactylon develops [34] and root biomass increases [35], indicating that the roots remain vital, enabling quick sprouting after the riparian zone is exposed. Additionally, during flooding, the aboveground stolons of C. dactylon quickly die off, while the underground rhizomes firmly anchor in the soil, absorbing soil nutrients and storing energy. When the aboveground stems of C. dactylon emerge from the water surface, photosynthesis and aerobic respiration can take place, facilitating rapid plant growth.

Plant height, upright stem length, primary stolon length, and root length of C. dactylon in the riparian zone were positively correlated with elevation and negatively correlated with flooding duration (Table 1). The dominance of C. dactylon in the 145~155 m elevation of the Three Gorges Reservoir riparian zone may be related to its internal aerenchyma tissue [36-38]. In the low elevation zone, the plant height and average internode length of C. dactylon decreased, whereas the number of tillers increased. This facilitated the rapid diffusion of gases from stems to roots, providing oxygen for root respiration and promoting root growth. Consequently, it increased tissue porosity and oxygen leakage, enhancing the porosity of primary and adventitious roots and the waterlogging tolerance of the plant. In the high elevation zone, as the duration of waterlogging decreased, the time available for photosynthesis and aerobic respiration increased, leading to more pronounced growth. Considering the growth pattern of C. dactylon under flooding conditions, shorter flooding durations allowed for a longer period of photosynthesis in the aboveground stems, resulting in longer vegetation growth. This aligns with the findings of this study.

Reviewer 5 Report

Comments - water-2546935-peer-review-v1

 Adaptive characteristics of stress-tolerant plants in the water- level fluctuation zone of Three Gorges Reservoir

The authors conducted an interesting work but falls under the scope of aquatic ecology. But I did not find any application or importance of the presented content of the article. Because does not have any direct link with the problems and management of water.  Yes, ecologically it has significant but lacks important points.

1.       Broad title and limited work.

2.       Title talks about the plants but the content presents Bermuda grass only. I think it should be Adaptive characteristics of Bermuda grass in the water level fluctuation zone of three Gorges reservoir.

3.       Abstract is written superficially lacks a single sentence for methods. Addresses Cyanodon dactylon. What about other plants???

4.       Introduction is better. Research gap and justification absent. Why this research is essential?? Any research question???

5.       Methods are ambiguous. Why transect of that size used … Why not quadrate?

6.       In the laid transects there must have been other species – what about them?

7.       Why was only Bermuda grass selected????

8.       The floristic composition is absent… Species, genus and families???

9.       What do you mean by Yellow and purple soil??? Need to be elucidated.

10.   How can you generalize the adaptive traits Bermuda grass for all plants?

11.   Only Bermuda was there???

12.   Why other species were not included in the study?

13.   In section 3.3 the plant name should be in italicized. What were the adaptive characteristics traits of the species mentioned here???

14.   There this need to increase the scale and extent of the study for further submission in any phytosociological or limnological journals.

--------------------------------------------------------------------------------------------------------------------------------

Quality of English language is good.

Author Response

The authors conducted an interesting work but falls under the scope of aquatic ecology. But I did not find any application or importance of the presented content of the article. Because does not have any direct link with the problems and management of water. Yes, ecologically it has significant but lacks important points.

  1. Broad title and limited work.

Response: Thank you very much for your comments. We have rewritten the title as What drives the morphological traits of stress-tolerant plant Cynodon dactylon in a riparian zone of the Three Gorges Reservoir, China”

  1. Title talks about the plants but the content presents Bermuda grass only. I think it should be Adaptive characteristics of Bermuda grass in the water level fluctuation zone of three Gorges reservoir.

Response: Thank you very much for your comments. We have rewritten the title as “What drives the morphological traits of stress-tolerant plant Cynodon dactylon in a riparian zone of the Three Gorges Reservoir, China”. We also added some description about the adaptive characteristics of Bermuda grass in the riparian zone of three Gorges reservoir. “Since C. dactylon is the most dominant plant species, covering almost all the riparian zones of the Three Gorges Reservoir, we selected it as our study object. The plant C. dactylon shows significant resistance to flooding and nutrient-poor conditions.

  1. Abstract is written superficially lacks a single sentence for methods. Addresses Cyanodon dactylon. What about other plants???

Response: Thank you very much for your comments. We have rewritten the title and abstract section. We also added some description about the methods and wrote some sentences addressed Cynodon dactylon in the abstract.

Abstract: The cyclical process of water storage and recession in the regular operation of the Three Gorges Reservoir creates a unique habitat stress that alters the structural and functional attributes of vegetation ecology within the riparian zone. The stress-tolerant plant Cynodon dactylon (L) Pers is the dominant plant species in the riparian zone of the Three Gorges Reservoir. In this study, the riparian zone of the Daning River, a tributary located central to the Three Gorges Reservoir, was selected as our study area. To identify the drivers of the morphological traits of C. dactylon in the riparian zone of Daning River, we examined plant biomass and plant characteristics across different elevation gradients, with reference to abiotic factors to determine the distribution patterns of plant morphological traits. Results indicated that in the two main soil types of the riparian zone, plant biomass showed a consistent trend along the elevation gradient, with a “middle-height expansion” pattern; biomass increased and then decreased with rising water levels. Plant biomass positively correlated with soil total nitrogen and negatively correlated with soil pH, electrical conductivity, and total phosphorus. C. dactylon adapted to prolonged flooding in the riparian zone by having a significant negative correlation between plant height and erect stem length with soil moisture content to facilitate root respiration.

  1. Introduction is better. Research gap and justification absent. Why this research is essential?? Any research question???

Response: Thank you very much for your comments. We have rewritten the introduction section by adding several sentences to describe the essential of this research.

The growth behavior and life history strategies of plants at each stage of their growth and development are closely related to the morphological traits of plants and can determine the distribution pattern and population behavior of plant species in the habitat to a certain extent. The plant life history strategy is the best resource allocation method for plant species to maintain growth and reproduction. The adaptability of species is accumulated in the process of species evolution, which is manifested in morphology. Different morphological growth strategies are adopted by plants to maximize the fitness of species at certain stages. Plant morphological traits play an important role in studying biogeography in many ecosystems such as forestry [12] and agricultural [13] ecosystems. Under drought stress, plants can obtain the water content required for normal growth and development through the plasticity of morphological and structural characteristics, and adapt to the stressed environment by changing their morphogenesis. However, the morphological traits of plants in the riparian zone of reservoirs have received little attention. Differences in abiotic factors caused by climate change and geographical location differences will lead to different morphological characteristics of vegetation [14, 15].

To maintain the C. dactylon community, it is important to address the morphological traits of this plant across a gradient of elevations, as well as its relationships with regional abiotic factors in the riparian zones. Hence, in this study, we tested the hypotheses that (1) C. dactylon morphological traits varied across elevations between two types of soil (yellow loam and purple soil) and (2) C. dactylon morphological traits varied with abiotic drivers.

  1. Methods are ambiguous. Why transect of that size used … Why not quadrate?

Response: Thank you very much for your comments. What we used is quadrate, now we rewrite the methods to make it easier for reader to understand.

  1. In the laid transects there must have been other species – what about them?

Response: The investigation results revealed that the dominant plant species within the 145 m to 165 m elevation range in the riparian zone of the Daning River were primarily C. dactylon, occasionally accompanied by A. macrocephala and H. aromatica. At elevations of 170 m and above, higher plant diversity was observed, including species such as Daucus carota L., Ambrosia artemisiifolia L., Conyza canadensis (L.) Cronq., Digitaria sanguinalis (L.) Scop., Bidens frondosa L., Vitex negundo L., Melilotus officinalis (L.) Desr., Beckmannia syzigachne (Steud.) Fern., and Setaria viridis (L.) Beauv. These species are mainly annual and perennial herbaceous plants. At higher elevations, shrubs, trees, and farmland were also present.

  1. Why was only Bermuda grass selected????

Response: Since C. dactylon is the most dominant plant species, covering almost all the riparian zones of the Three Gorges Reservoir, we selected it as our study object. The plant C. dactylon shows significant resistance to flooding and nutrient-poor conditions [8,9].

  1. The floristic composition is absent… Species, genus and families???

Response: We have showed the detailed species name.

  1. What do you mean by Yellow and purple soil??? Need to be elucidated.

Response: Purple soil and yellow loam were the broadest soil types in the riparian zone of the Three Gorges Reservoir [16].

  1. How can you generalize the adaptive traits Bermuda grass for all plants?

Response: Since C. dactylon is the most dominant plant species, covering almost all the riparian zones of the Three Gorges Reservoir. We have also rewritten the title as What drives the morphological traits of stress-tolerant plant Cynodon dactylon in a riparian zone of the Three Gorges Reservoir, China”

  1. Only Bermuda was there???

Response: The investigation results revealed that the dominant plant species within the 145 m to 165 m elevation range in the riparian zone of the Daning River were primarily C. dactylon, occasionally accompanied by A. macrocephala and H. aromatica. At elevations of 170 m and above, higher plant diversity was observed, including species such as Daucus carota L., Ambrosia artemisiifolia L., Conyza canadensis (L.) Cronq., Digitaria sanguinalis (L.) Scop., Bidens frondosa L., Vitex negundo L., Melilotus officinalis (L.) Desr., Beckmannia syzigachne (Steud.) Fern., and Setaria viridis (L.) Beauv. These species are mainly annual and perennial herbaceous plants. At higher elevations, shrubs, trees, and farmland were also present.

  1. Why other species were not included in the study?

Response: The investigation results revealed that the dominant plant species within the 145 m to 165 m elevation range in the riparian zone of the Daning River were primarily C. dactylon, occasionally accompanied by A. macrocephala and H. aromatica. At elevations of 170 m and above, higher plant diversity was observed, including species such as Daucus carota L., Ambrosia artemisiifolia L., Conyza canadensis (L.) Cronq., Digitaria sanguinalis (L.) Scop., Bidens frondosa L., Vitex negundo L., Melilotus officinalis (L.) Desr., Beckmannia syzigachne (Steud.) Fern., and Setaria viridis (L.) Beauv. These species are mainly annual and perennial herbaceous plants. At higher elevations, shrubs, trees, and farmland were also present.

Since C. dactylon is the most dominant plant species, covering almost all the riparian zones of the Three Gorges Reservoir. We have also rewritten the title as What drives the morphological traits of stress-tolerant plant Cynodon dactylon in a riparian zone of the Three Gorges Reservoir, China”

  1. In section 3.3 the plant name should be in italicized. What were the adaptive characteristics traits of the species mentioned here???

Response: Thank you, we have italicized the plant name. Results indicated that in the two main soil types of the riparian zone, plant biomass showed a consistent trend along the elevation gradient, with a “middle-height expansion” pattern; biomass increased and then decreased with rising water levels. Plant biomass positively correlated with soil total nitrogen and negatively correlated with soil pH, electrical conductivity, and total phosphorus. C. dactylon adapted to prolonged flooding in the riparian zone by having a significant negative correlation between plant height and erect stem length with soil moisture content to facilitate root respiration.

  1. There this need to increase the scale and extent of the study for further submission in any phytosociological or limnological journals.

Response: Thank you, we have revised the whole manuscript to meet the journal Water.

Round 2

Reviewer 4 Report

Manuscript “ Adaptive characteristics of stress-tolerant plants in the water- level fluctuation zone of Three Gorges Reservoir”, authors Xiaolong Li , Shanze Li, Yawei Xie, Zehui Wei and Zilong Li

 Introduction

L76-77 The following statement is unclear: The plant life history strategy is the best resource 76 allocation method for plant species to maintain growth and reproduction.

  Results

 L 184 …bars sharing a letter are not  significantly different from one another…Do you mean bars sharing the same letter are not  significantly different from one another?

 L 232-233  Relationship between riparian plant biomass and C. Dactylon morphological 232 traits with environmental factors…Write “Dactylon” with low capital letter

 N as a number under the figures are sometimes written with small letter and sometime with capital letter! Make it uniform, please!

 My suggestion: minor revision

Author Response

Introduction

L76-77 The following statement is unclear: The plant life history strategy is the best resource 76 allocation method for plant species to maintain growth and reproduction.

Response: Since this sentence is far away from our topic, we deleted it.

  Results

 L 184 …bars sharing a letter are not significantly different from one another…Do you mean bars sharing the same letter are not significantly different from one another?

Response: Yes. To make it clear, we have revised the sentence as “bars sharing the same letter are not significantly different from one another”.

 L 232-233 Relationship between riparian plant biomass and C. Dactylon morphological 232 traits with environmental factors…Write “Dactylon” with low capital letter

Response: Thank you for your careful review, we have revised the “Dactylon” with low capital letter.

 N as a number under the figures are sometimes written with small letter and sometime with capital letter! Make it uniform, please!

Response: Thank you for your careful review, we have uniformed the N with capital letter.

Reviewer 5 Report

The MS is fine now for possible publication in the Journal.

Author Response

Thank you for your comments.